# The association between walking speed and risk of cardiovascular disease in middle-aged and elderly people in Taiwan, a community-based, cross-sectional study

Yu-Lin Shih[1‡], Chin-Chuan Shih[2], Jau-Yuan Chen[1,3‡]*

1 Department of Family Medicine, Chang-Gung Memorial Hospital, Linkou Branch, Taoyuan City, Taiwan (R. O.C.), 2 General Administrative Department, United Safety Medical Group, New Taipei City, Taiwan (R.O. C.), 3 Chang Gung University College of Medicine, Taoyuan City, Taiwan (R.O.C.)

‡ Co-first authors on this work.
* welins@cgmh.org.tw

**Data Availability Statement:** Our research is based on the use and analysis of anonymized data by the Chang-Gung Medical Foundation Institutional Review Board. The original data cannot

## Abstract

### Background and aims

The aim of this study was to investigate the association between walking speed and cardio-vascular disease (CVD) risk among community-dwelling middle-aged and elderly populations in Taiwan.

### Methods

This was a cross-sectional and community-based study with 400 participants aged 50 years and over recruited from a community health promotion project in 2014 in Guishan district, Taoyuan city. We excluded 91 people, and a total of 309 participants were eligible for analysis. The statistical methods used in this study were one-way ANOVA and the Chi-square test, Pearson's correlation test and logistic regression model.

### Results

In total, 309 participants (98 males and 211 females) aged 50 to 74 (62.05 ± 6.21) years without a CVD history were enrolled in this study. The walking speed gradually decreased from the low CVD risk group to the high CVD risk group (p < 0.05). A significant inverse association between walking speed and CVD risk was confirmed with a Pearson's correlation coefficient of -0.143 (p < 0.05) in middle-aged people, but this significant inverse association was not shown in elderly people. The multivariate logistic regression model for predicting CVD risk and walking speed with an adjusted odds ratio (OR) was 0.127 (95% CI = 0.021–0.771) in middle-aged people with adjustment for sex, age, waist circumference (WC), hypertension (HTN), diabetes mellitus (DM), and hyperlipidemia (p < 0.05).

be shared here due to ethical restrictions regarding potentially identifying or sensitive information. Due to ethical restrictions imposed by the the Chang-Gung Medical Foundation Institutional Review Board, the subjects data are not available for distribution outside of the Chang-Gung Memorial Hospital. Data access requests can be submitted to CHUN-MING, Study Manager and Data Coordinator, Email: fm33h00@gmail.com Phone: 886-33196200-3413.

**Funding:** This study was supported by Chang Gung Memorial Hospital (grants CORPG3C0171~3C0172, CZRPG3C0053, CORPG3G0021, CORPG3G0022).

**Competing interests:** The authors have declared that no competing interests exist.

## Conclusion

Our study clearly shows that slow walking speed is associated with an increased risk of CVD in middle-aged people rather than in elderly people.

## Introduction

According to the National Center for Health Statistics, heart disease is the leading cause of death in the United States [1]. The World Health Organization has also indicated that cardiovascular disease (CVD) is the number one cause of death globally [2], causing great loss and heavy burden in society [3]. There have been many attempts to predict CVD risk [4], and the Framingham Heart Study has been the most famous system for prediction [5]. Physical activity has been shown to reduce CVD risk in both women and men [6–8], and walking speed could be one of the simple and reliable ways to predict CVD risk [9]. Regular exercise has also been recognized as an important factor in reducing the Framingham risk score in patients with metabolic syndrome [10–12]. Physical activity could serve as a protective factor and an indicator for CVD. However, previous research has mainly focused on the relationship between walking speed and CVD risk only. Our research was the first community-based study that aimed to reveal the role of age in the relationship of walking speed and CVD risk.

## Materials and methods

### Study design and participants

This was a cross-sectional and community-based study. Before this study, the minimum sample size for this study was calculated at the initial stage of the study after previewing a relatively smaller population. Considering an 85% power, a 95% confidence level, and 0.30 as the high CVD risk rate among middle-aged people, we calculated that 288 participants were required to detect differences between these two study groups with an odds ratio of at least 2. We recruited 400 participants aged 50 years and over from a community health promotion project in 2014 in Guishan district, Taoyuan city. We excluded 91 participants who (1) had a history of CVD, (2) were unable to complete a 6-meter-long walk, and (3) were over 75 years old (Fig 1). A total of 309 participants were eligible for analysis and were divided into three groups based on three levels of CVD risk: low, medium, and high. Each participant completed a questionnaire that included personal information and a medical history during a face-to-face interview. Furthermore, they completed a laboratory measurement and a 6-meter walking speed test. The study was approved by the Chang-Gung Medical Foundation Institutional Review Board (102-2304B), and all the participants were fully informed and signed informed consent forms before enrollment.

### Data collection and laboratory measurements

The content of the questionnaire included sex, age, alcohol drinking status (drinking ≥2 days/week or not) and current smoking status (current smoker or not). The health survey contained hypertension (HTN), diabetes mellitus (DM), and dyslipidemia data. Resting systolic and diastolic blood pressure (BP, mmHg) were measured at least two times at rest. The biochemical laboratory data were analyzed at the central laboratory of Linkou Chang Gung Memorial Hospital, including low density lipoprotein (LDL-C, mg/dl), total cholesterol (TC, mg/dl), high-

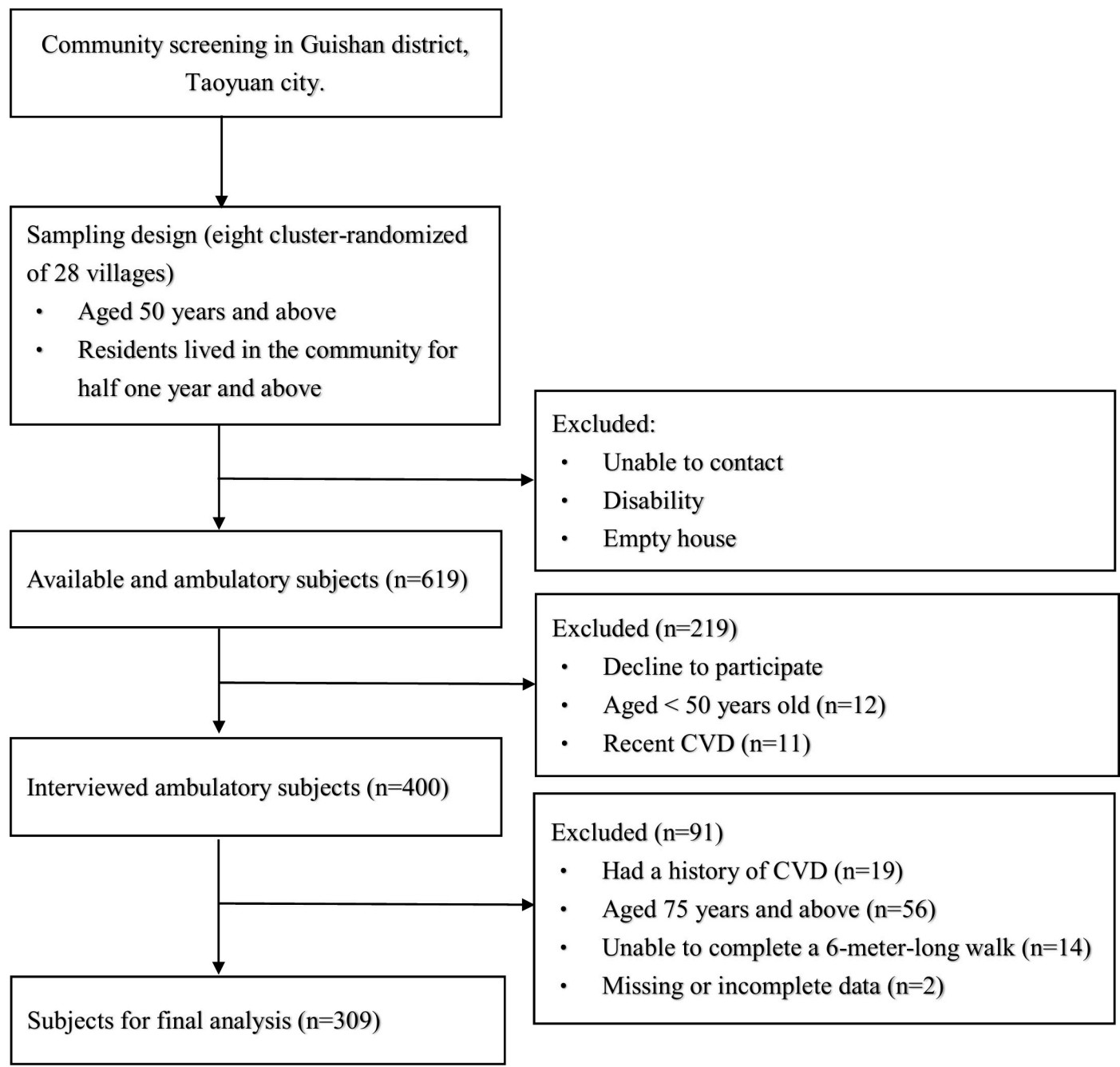

**Fig 1. Flow chart of study subjects.**

density lipoprotein (HDL-C, mg/dl), triglyceride (TG, mg/dl), waist circumference (WC, cm), alanine transaminase (ALT, mg/dl), creatinine (mg/dl), fasting plasma glucose (FPG, mg/dl), and uric acid (mg/dl). WC was measured at the midpoint between the inferior margin of the last rib and the iliac crest in a horizontal plane in an upright position. We also measured the time (in seconds) it took for participants to walk 6 meters in a straight line to obtain the walking speed data.

## Definition of CVD risk

In this study, the general CVD risk model from the Framingham Heart Study was used for predicting CVD risk, which included several variables, such as age, sex, systolic BP, anti-HTN medication, current smoking status, DM, TC, and HDL-C [13]. DM was defined as an FPG ≥ 126 mg/dL or the use of oral hypoglycemic agents or insulin therapy. HTN was defined as systolic BP ≥ 140 mmHg, diastolic BP ≥ 90 mmHg, or the use of treatment for HTN.Dyslipidemia was defined as LDL-C ≥ 130 mg/dL, HDL-C < 40 mg/dL in males or < 50 mg/dL in females, TG ≥ 150 mg/dL, TC ≥ 200 mg/dL, or the use of lipid-lowering medication. Current smoking status was recorded by self-report. We further divided our participants into a low CVD risk group (Framingham Risk Score [FRS] < 10%), a medium-risk group (10% ≤ FRS < 20%), and a high-risk group (FRS ≥ 20%).

## Statistical analysis

Participants were divided into three groups according to three levels of CVD risk: low, medium, and high. For the laboratory and clinical data of each group, continuous variables were expressed as the mean ± SD and were analyzed by one-way ANOVA. Categorical variables were expressed as n (%) and analyzed with a chi-square test. In addition, participants were divided into a middle-aged group and an elderly group according to age. Pearson's correlation coefficient was used to analyze correlations between walking speed and TC, HDL, LDL, TG, FPG, uric acid, WC, and FRS scores in each group. Finally, the multivariate logistic regression model for predicting CVD risk and walking speed was adjusted for sex, age, WC, HTN, DM, and hyperlipidemia in each group. In our study, a p-value of < 0.05 was considered statistically significant. All statistical analyses were performed using SPSS for Windows (IBM Corp. Released 2011. IBM SPSS Statistics, version 20.0. Armonk, NY: IBM Corp.)

## Results

The 309 participants were 50-74(62.05 ± 6.21) years old and consisted of 98 males (31.7%) and 211 females (68.3%) with no CVD history. The general characteristics of the study participants are shown in Table 1. There were significantly more males than females with a high CVD risk, and there were significant increases in age, current smoking status, HTN, DM, SBP, DBP, and HDL-C in the group with a higher CVD risk. Other parameters that also increased with higher CVD risk were TG, WC, creatinine, FPG, and uric acid. There was no significant difference in LDL-C or TC among the three groups. The walking speeds of the low, medium and high CVD risk groups were 1.05 ± 0.31, 0.98 ± 0.24, and 0.94 ± 0.28, respectively. The differences among the three groups were statistically significant. The walking speed gradually decreased from the low CVD risk group to the high CVD risk group (p < 0.05). In Table 2 and Table 3, participants were divided into two groups according to age. Those older than or equal to 50 years old and younger than 65 years old were in the middle-aged group, and those older than 65 years old were in the elderly group.

In Table 2, the walking speed of the middle-aged group showed negative relationships with LDL, TG, FPG, uric acid, and CVD risk but showed positive relationships with TC and HDL. Only the relationships between walking speed and HDL and CVD risk reached statistical significance in middle-aged people. On the other hand, the walking speed of the elderly group showed negative relationships with TC, HDL, LDL, TG, FPG, uric acid, and CVD risk but showed a positive relationship with WC. Only the relationship between walking speed and uric acid reached statistical significance in elderly people. A significant inverse association between walking speed and FSR was confirmed with a Pearson's correlation coefficient of -0.143 (p < 0.05) in middle-aged people rather than in elderly people.

**Table 1. The characteristics of study subjects categorized by CVD risk.**

| Variables | Low risk (n = 139) | Moderate risk (n = 99) | High risk (n = 71) | p value |
|---|---|---|---|---|
| Sex (male), n(%) | 11 (7.9) | 33 (33.3) | 54 (76.1) | <0.001 |
| Age (years) | 59.1 ± 5.4 | 64.7 ± 6.1 | 64.1 ± 5.4 | <0.001 |
| Alcohol drinking, n(%) | 20 (14.4) | 18 (18.2) | 21 (29.6) | 0.029 |
| Current Smoking, n(%) | 2 (14.4) | 6 (6.1) | 23 (32.4) | <0.001 |
| Hypertension, n(%) | 34 (24.5) | 60 (60.6) | 53 (74.6) | <0.001 |
| Diabetes, n(%) | 12 (8.6) | 15 (15.2) | 28 (39.4) | <0.001 |
| Dyslipidemia, n(%) | 78 (56.1) | 67 (67.7) | 54 (76.1) | 0.012 |
| walking speed (m/s) | 1.05 ± 0.31 | 0.98 ± 0.24 | 0.94 ± 0.28 | 0.03 |
| WC (cm) | 81.1 ± 7.7 | 84.5 ± 8.4 | 91.0 ± 10.2 | <0.001 |
| SBP (mmHg) | 121.3 ± 12.6 | 132.6 ± 14.6 | 138.1 ± 17.9 | <0.001 |
| DBP (mmHg) | 74.4 ± 9.4 | 79.1 ± 10.9 | 84.1 ± 11.1 | <0.001 |
| LDL-C (mg/dl) | 116.3 ± 30.5 | 123.8 ± 33.2 | 120.1 ±33.3 | 0.209 |
| TC (mg/dl) | 197.3 ± 33.3 | 203.4 ± 36.3 | 196.4 ± 36.7 | 0.324 |
| HDL-C (mg/dl) | 60.3 ± 13.3 | 54.6 ± 13.1 | 45.9 ± 9.5 | <0.001 |
| TG (mg/dl) | 103.1 ± 47.2 | 125.6 ± 60.8 | 152.4 ± 78.0 | <0.001 |
| ALT (mg/dl) | 21.1 ± 8.8 | 22.4 ± 13.5 | 27.0 ± 17.4 | 0.006 |
| Creatinine (mg/dl) | 0.65 ± 0.27 | 0.74 ± 0.21 | 0.86 ± 0.37 | <0.001 |
| FPG (mg/dl) | 90.0 ± 12.6 | 96.7 ± 24.3 | 105.7 ± 29.2 | <0.001 |
| Uric acid (mg/dl) | 5.4 ± 1.1 | 5.7 ± 1.4 | 6.4 ± 1.7 | <0.001 |
| CVD risk (%) | 6.2 ± 2.2 | 14.0 ± 2.6 | 30.2 ± 11.8 | <0.001 |

(low risk group:FRS< 10%, moderate risk group:10% ≤ FRS <20%, high risk group:FRS ≥ 20%; FRS = Framingham Risk Score).

(Data expressed as mean ± SD for continuous variables and n (%) for categorical variables).

(Data analyzed with One-way ANOVA for continuous variables and chi-squared test for categorical variable).

(The definition of low, moderate and high risk are FRS < 10%, 10% ≤ FRS <20% and FRS ≥ 20% respectively).

Abbreviations: SBP = systolic blood pressure; DBP = diastolic blood pressure; TC = total cholesterol; LDL-C = low density lipoprotein; HDL-C = high density lipoprotein; TG = triglyceride; WC = waist circumference; ALT = alanine transaminase; FPG = fasting plasma glucose; FRS score = Framingham Risk Score.

**Table 2. Pearson's correlation coefficients (r) between walking speed and CVD risk factors.**

| Variables | Walking speed (m/s) | |
|---|---|---|
| | r (middle-aged) | r (elderly) |
| TC (mg/dl) | 0.036 | -0.117 |
| HDL (mg/dl) | 0.143* | -0.065 |
| LDL (mg/dl) | -0.074 | -0.132 |
| TG (mg/dl) | -0.089 | -0.077 |
| FPG (mg/dl) | -0.009 | -0.062 |
| Uric acid (mg/dl) | -0.045 | -0.219* |
| WC (cm) | -0.004 | 0.014 |
| CVD risk (%) | -0.143* | -0.032 |

r = correlation coefficients,

* = p value < 0.05.

Abbreviations: TC = total cholesterol; LDL-C = low density lipoprotein; HDL-C = high density lipoprotein; TG = triglyceride; FPG = fasting plasma glucose; WC = waist circumference; CVD risk was measured by Framingham Risk Score (FRS).

**Table 3. Odds Ratio (OR) for moderate to high CVD risk according to walking speed.**

| Walking speed | Middle-aged (n = 209) | | Elderly (n = 100) | |
|---|---|---|---|---|
| | OR (95% C.I.) | p | OR (95% C.I.) | p |
| Model 1 | 0.346 (0.125–0.961) | 0.046 | 2.523 (0.363–17.524) | 0.349 |
| Model 2 | 0.276 (0.078–0.984) | 0.047 | 3.329 (0.321–34.540) | 0.314 |
| Model 3 | 0.127 (0.021–0.771) | 0.025 | 27.259 (0.752–987.664) | 0.071 |

(data expressed as OR and 95% confidence interval(CI)).

Odds Ratios for walking speed by FRS score, separating to FRS < 10% and FRS≥10% two groups.

†Model 1: unadjusted; Model 2: adjusted for sex and age; Model 3: adjusted for sex, age, waist circumference, hypertension, diabetes mellitus and hyperlipidemia.

In Table 3, three multiple logistic regression models were used to calculate the OR of walking speed with CVD risk under adjustment for other risk factors. Model 1 was unadjusted; model 2 was adjusted for sex and age; and model 3 was adjusted for sex, age, WC, HTN, DM, and hyperlipidemia. The p-values of both groups decreased when considering more factors in the multiple logistic regression model, and only the p-values of the middle-aged group remained statistically significant in all three models. In particular, in model 3, the p-value of the middle-aged group reached 0.025 as the minimum value in Table 3, indicating the strongest relationship between walking speed and CVD risk. The multivariate logistic regression model for predicting CVD risk and walking speed showed an adjusted OR of 0.127 (95% CI = 0.021–0.771) in middle-aged people with adjustment for sex, age, WC, HTN, DM, and hyperlipidemia. It also confirmed the strongest relationship between walking speed and CVD risk in middle-aged people ($p < 0.05$) rather than in elderly people.

## Discussion

In this community-based cross-sectional study, we aimed to discuss the association between walking speed and CVD risk in people older than 50 years of age, and we found evidence that the risk of CVD was highly related to walking speed in people between the ages of 50 and 65 but not in people over 65 years of age.

In Table 1, we used FRS as a CVD risk measurement to categorize participants into three groups: the low CVD risk group (FRS < 10%), medium-risk group (10% ≤ FRS < 20%), and high-risk group (FRS ≥ 20%). We compared laboratory data and all participants' lifestyles in Table 1. The risk factors for cardiovascular risk were sex, age, systolic pressure, HTN, current smoking status, DM, HDL-C and TC, as shown in Table 1. Men had a higher CVD risk than women, and this result corresponds to the fact that men have a higher cardiovascular risk [14]. Old age is another important risk factor for CVD [15], and our study showed that older people had a higher CVD risk. HTN and high systolic pressure also play important roles in CVD. High BP usually elevates CVD risk and increases CVD morbidity [16], and high systolic pressure was positively correlated with high CVD risk in our study. Smoking-induced vessel damage is also associated with CVD [17], and DM is another risk factor for myocardial ischemia and CVD [18], both of which were related to elevated CVD risk in this study. On the other hand, HDL-C is a protective factor for CVD [19]. Therefore, this study showed that a lower CVD risk tended to result in a higher level of HDL-C. Although high cholesterol levels are linked to elevated CVD risk [20], there was no significant difference found in our study, as shown in Table 1.

Table 1 shows a significant difference in the group with high FRS with a lower walking speed (p = 0.03). Table 2 focuses on the relationship between age and walking speed and shows

**Table 4. Comparison with the other recent epidemiological studies in the similar field.**

| Title | Author | Date | Study design | Finding |
|---|---|---|---|---|
| Slow walking speed and cardiovascular death in well functioning older adults: prospective cohort study [25] | Julien Dumurgier.etc | 2009 | prospective cohort study | lower walking speed was strongly associated with cardiovascular mortality in a populationof well functioning older people. |
| Association of walking speed in late midlife with mortality: results from the Whitehall II cohort study [26] | Alexis Elbaz.etc | 2013 | cohort study | slow walking speed during late midlife is associated with increased mortality over 6 years of follow-up |
| Objective measures of the frailty syndrome (hand grip strength and gait speed) and cardiovascular mortality: A systematic review [27] | Vinod Chainani.etc | 2016 | A systematic review | both gait speed and hand grip strength are associated with increased risk of cardiovascular mortality in diverse populations. |
| Treadmill walking speed and survival prediction in men with cardiovascular disease: a 10-year follow-up study [28] | Giorgio Chiaranda.etc | 2013 | Population-based prospective study | 1 km treadmill walking test at a moderate intensity is a useful tool for identifying mortality risk in patients with stable cardiovascular disease. |

a strong negative relationship. Participants over 50 years old were divided into two groups: those older than or equal to 50 years old but below 65 years old were in the middle-aged group, and those older than 65 years old were in the elderly group. The walking speed of the two groups was negatively correlated with LDL, TG, FPG, uric acid, and CVD risk; only the walking speed of the middle-aged group showed a positive correlation and reached statistical significance. This result indicates that walking speed is closely related to CVD risk in middle-aged people rather than in elderly people.

There are many factors that can weaken the physical activity of middle-aged and elderly people. The models were adjusted for all the factors listed in Table 3 to analyze their impact on the relationship between walking speed and CVD risk. There were three models to adjust the ORs between walking speed and CVD risk. Model 1 was unadjusted, and the ORs of the middle-aged group and the elderly group were 0.246 and 2.523, respectively, but only the OR of the middle-aged group reached statistical significance, showing that walking speed is highly associated with CVD risk in middle-aged people. Model 2 was adjusted for age and sex, and model 3 was adjusted for sex, age, WC, HTN, DM, and hyperlipidemia. The OR in the middle-aged group was statistically significant in both model 2 and model 3, and the p-value decreased when more factors were adjusted in model 3, indicating that walking speed is a powerful indicator of CVD risk in the middle-aged group. On the other hand, the OR never reached statistical significance in the models of elderly people, although the p-value decreased when more factors were adjusted, indicating that there are more potential factors that can affect the relationship of walking speed and CVD risk. There are some physical challenges related to advanced age that can be potential factors affecting walking speed, such as arthritis, chronic obstructive pulmonary disease, and sarcopenia [21–24]. These physical challenges may explain why the elderly group did not show a strong negative relationship between walking speed and CVD risk. However, we did not discuss the potential factors in this study. Therefore, this study indicates that walking speed alone can be a predictor of CVD risk in the middle-aged population, but for the elderly population, there are more factors to be considered. In Table 4, previous research has mainly focused on the relationship between walking speed and CVD risk [25–28] in other countries. Our research was the first community-based study that examined the role of age in the relationship between walking speed and CVD risk in Taiwan. Thus, the novel findings of this study are the first to explore the association between CVD risk and walking speed in different age populations in Taiwan. Second, we conducted a community-based study and comprehensively collected various data from a health promotion project, which may have clinical implications.

However, our study still had limitations. The participants in our study were recruited from a relatively small group in northern Taiwan as our favored population. The characteristics of

the residents from this small group might differ from characteristics in the general population, so the findings cannot be generalized to the whole middle-aged and elderly population in Taiwan. Selection bias should be considered, and the results of our study should not be extrapolated to other regions of Taiwan. Future studies using random sampling of communities with a wider range of regions would make the research more discursive.

## Conclusion

Our study shows that slow walking speed is highly associated with high CVD risk in middle-aged people. However, as discussed above, there are more factors that lead to a decline in exercise capacity or an increased risk of CVD in elderly individuals. In conclusion, using walking speed alone to predict CVD risk can only be used in middle-aged people because there are more factors to be considered in elderly people. Thus, our findings may provide valuable information for clinicians to alert middle-aged people regarding the increased risk of CVD.

## Author Contributions

**Conceptualization:** Yu-Lin Shih.

**Data curation:** Jau-Yuan Chen.

**Formal analysis:** Chin-Chuan Shih.

**Methodology:** Jau-Yuan Chen.

**Project administration:** Jau-Yuan Chen.

**Writing – original draft:** Yu-Lin Shih.

**Writing – review & editing:** Yu-Lin Shih.

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
