## [Decision Letter · Decision Letter 0]

16 Mar 2020

PONE-D-20-04269

The Association between Walking Speed and Risk of Cardiovascular Disease in Middle-aged and Elderly people

PLOS ONE

Dear Dr. Chen,

Thank you for submitting your manuscript to PLOS ONE. After careful consideration, we feel that it has merit but does not fully meet PLOS ONE’s publication criteria as it currently stands. Therefore, we invite you to submit a revised version of the manuscript that addresses the points raised during the review process.

Both reviewers raised several concerns that need to be addressed. Particularly, one reviewer identified language problems in your manuscript. Your revision must be professionally edited to improve the language. You also need to upload the certificate of this English editing service along with your revision.

We would appreciate receiving your revised manuscript by Apr 30 2020 11:59PM. To enhance the reproducibility of your results, we recommend that if applicable you deposit your laboratory protocols in protocols.io, where a protocol can be assigned its own identifier (DOI) such that it can be cited independently in the future. For instructions see: http://journals.plos.org/plosone/s/submission-guidelines#loc-laboratory-protocols

We look forward to receiving your revised manuscript.

Kind regards,

Yu Ru Kou, PhD

Academic Editor

PLOS ONE

Journal Requirements:

2. Your ethics statement must appear in the Methods section of your manuscript. If your ethics statement is written in any section besides the Methods, please move it to the Methods section and delete it from any other section. Please also ensure that your ethics statement is included in your manuscript, as the ethics section of your online submission will not be published alongside your manuscript.

Reviewers' comments:

Reviewer's Responses to Questions

**Comments to the Author**

1. Is the manuscript technically sound, and do the data support the conclusions?

Reviewer #1: Partly

Reviewer #2: Partly

2. Has the statistical analysis been performed appropriately and rigorously? 

Reviewer #1: Yes

Reviewer #2: Yes

3. Have the authors made all data underlying the findings in their manuscript fully available?

Reviewer #1: Yes

Reviewer #2: Yes

4. Is the manuscript presented in an intelligible fashion and written in standard English?

Reviewer #1: Yes

Reviewer #2: No

5. Review Comments to the Author

Reviewer #1: The Authors propose the community-based study to explore the association between walking speed and cardiovascular disease (CVD) risk among community-dwelling middle-aged and elderly population in Taiwan. The study designs and methods used are basically appropriate, and the interpretations of the results are reasonable. However, there are several areas where the manuscript needs to be strengthened.

1.Please give the power of data collection.

2.Flow chart of selection of the study sample and procedure is suggested.

3.A statement including the reference number of the ethics committee where appropriate should appear in the manuscript.

4.From the epidemiologic viewpoint, there are many confounding factors in the evidenced-based researches. How the authors deal with associated confounding factors in this study?

5.The authors should point and clarify the feature and novel findings of this study.

6.Please consider the comparison with the other epidemiological studies in other areas using table so make clear the significance of this study.

7.The results of goodness-of-test in table 3 are suggested.

8.The authors should add the comments related to selection bias in this study to the perceived limitation subsection.

9.More discussion regarding the policy implications of their findings would be important for the use of methodology in health policy making.

10.Some references should be updated.

Totally, I would like to congratulate the authors for the enthusiasm invested in this study. However, the manuscript does not reach the level of quality required for publication as original article without major revision in PLOS ONE.

Reviewer #2: The authors aimed to evaluate the walking speed for predicting cardiovascular disease (CVD) risk in middle-aged and elderly people. I am interested in this topic, however, the data is mainly collected from a specific community. Why did the authors choose the residents living in Guishan district to be enrolled in their study? Can this small group represent and draw conclusion in whole middle-aged and elderly population of Taiwan including the downtown, suburban, and rural areas?

General Comments:

I applaud the authors for their effort, however, the syntax, grammar and overall style of this manuscript is sometimes difficult to read and needs a robust editing from a fluent or native English speaker. For instance, in Introduction section, last sentence: “…and elderly poeple.” is an obvious typo error. I suggest the authors to have the manuscript carefully proofread again to correct grammar and other errors.

Major Comment 1:

Please revise the title of your manuscript so that it contains details of the study design which characterize the investigation as well. Please provide more information on the novelty of your study. Introduction should finish with the aim of the study that should be described more clearly.

Major Comment 2:

There were total 400 participants in the project, which was a relatively small sample size. Sample size should be increased to claim scientific merits of your study. The claims made by authors are significantly weakened by the fact that they are talking about findings drawn from a small sample size, and then should have mentioned this as limitation without any justification. Did you calculate the sample size before the investigation and what should be the minimal sample size needed in this study to avoid underpowered? Could the authors show the power of their sample size?

Major Comment 3:

Please indicate the selection of study participants by adding a CONSORT diagram with flow chart.

Major Comment 4:

The Introduction and Discussion sections need to be expanded with improvements. Try to cite more references to support your Result. Moreover, some of the references are outdated. Please update your reference list.

6. PLOS authors have the option to publish the peer review history of their article (what does this mean?). If published, this will include your full peer review and any attached files.

Reviewer #1: No

Reviewer #2: No

---

## [Author Response · Author response to Decision Letter 0]

29 May 2020

Responses to reviewers’ comments

Reviewer 1

Comment#1

Please give the power of data collection.

Response#:1

We thank the reviewer for reminding us this important issue. The sample size determination was based on the G*power 3.1 software. We did calculate the minimum sample size before this study. We also added statements to explain the calculation in study design and participants. These statements read as: “Before this study, the minimum sample size for this study was calculated at the initial stage of the study after previewing a relatively smaller population. Considering an 85% power, a 95% confidence level, and 0.30 as the high CVD risk rate among middle-aged people, we calculated that 288 participants were required to detect differences between these two study groups with an odds ratio of at least 2.” (Lines 79-83).

1. Jacob C, Statistical power analysis for the behavioral sciences. International Biometric Society, 1988.

Comment#2

Flow chart of selection of the study sample and procedure is suggested

Response#:2

Thanks for the reviewer’s suggestion, we made the flow chart in response to this comment (Please see Appendix 1), and put it in manuscript as Figure 1.

Comment#3

A statement including the reference number of the ethics committee where appropriate should appear in the manuscript.

Response#:3

We thank the reviewer for reminding us this important issue. In response to this comment, the reference number is 102-2304B and we add it into our manuscript in study design and participants, which read as” The study was approved by the Chang-Gung Medical Foundation Institutional Review Board (102-2304B), and all the participants were fully informed and signed informed consent forms before enrollment.” (Lines 90-92)

Comment#4

From the epidemiologic viewpoint, there are many confounding factors in the evidenced-based researches. How the authors deal with associated confounding factors in this study?

Response#:4

Thanks for the reviewer’s viewpoint. We deal those confounding factor with logistic regression in Table 3. Sex, age, waist circumference, hypertension, diabetes mellitus and hyperlipidemia were used for adjustment. We add material in statistical analysis, which read as “Finally, the multivariate logistic regression model for predicting CVD risk and walking speed was adjusted for sex, age, WC, HTN, DM, and hyperlipidemia in each group.” (lines 154-155), and we add another material in result, which read as “The multivariate logistic regression model for predicting CVD risk and walking speed with an adjusted odds ratio (OR) was 0.127 (95% CI = 0.021-0.771) in middle-aged people with adjustment for sex, age, waist circumference (WC), hypertension (HTN), diabetes mellitus (DM), and hyperlipidemia (p < 0.05).” (lines 53-57) 

Comment#5

The authors should point and clarify the feature and novel findings of this study.

Response#:5

We fully agree with the reviewer’s comment about novelty finding of our study, we have added statements in discussion, which read as: “Previous research has mainly focused on the relationship between walking speed and CVD risk [25] [26] [27] [28] in other countries. Our research was the first community-based study that examined the role of age in the relationship between walking speed and CVD risk in Taiwan. Thus, the novel findings of this study are the first to explore the association between CVD risk and walking speed in different age populations in Taiwan. Second, we conducted a community-based study and comprehensively collected various data from a health promotion project, which may have clinical implications.” (Lines 299-305).

Comment#6

Please consider the comparison with the other epidemiological studies in other areas using table so make clear the significance of this study.

Response#:6

We appreciate the reviewer’s comments and made a chart to compare with recent publication, and put it in manuscript as Table 4:

title author date Study design finding

Slow walking speed and cardiovascular death in well functioning older adults: prospective cohort study Julien Dumurgier.etc 2009 prospective cohort study lower walking speed was strongly associated with cardiovascular mortality in a population of well functioning older people.

Association of walking speed in late midlife with mortality: results from the Whitehall II cohort study Alexis Elbaz.etc 2013 cohort study slow walking speed during late midlife is associated with increased mortality over 6 years of follow-up 

Objective measures of the frailty syndrome (hand grip strength and gait speed) and cardiovascular mortality: A systematic review Vinod Chainani.etc 2016 A systematic review both gait speed and handgrip strength are associated with increased risk of cardiovascular mortality in diverse populations.

Treadmill walking speed and survival prediction in men with cardiovascular disease: a 10-year follow-up study Giorgio Chiaranda.etc 2013 Population-based prospective study 1 km treadmill walking test at a moderate intensity is a useful tool for identifying mortality risk in patients with stable cardiovascular disease.

Comment#7

The results of goodness-of-test in table 3 are suggested.

Response#:7

We appreciate the suggestion from reviewer. According to the p-value for hosmer and lemeshow goodness-of-fit test＞0.05

Comment#8

The authors should add the comments related to selection bias in this study to the perceived limitation subsection.

Response#:8

We fully agree with the reviewer regarding the fact that our findings were obtained from community-based subjects and cannot be generalized to the whole middle-aged and elderly population in Taiwan. To response to the reviewer’s comment, this limitation has been acknowledged in the discussion section, which read as “However, our study still had limitations. The participants in our study were recruited from a relatively small group in northern Taiwan as our favored population. The characteristics of the residents from this small group might differ from characteristics in the general population, so the findings cannot be generalized to the whole middle-aged and elderly population in Taiwan. Selection bias should be considered, and the results of our study should not be extrapolated to other regions of Taiwan. Future studies using random sampling of communities with a wider range of regions would make the research more discursive.” (Lines 306-312).

Comment#9

More discussion regarding the policy implications of their findings would be important for the use of methodology in health policy making.

Response#:9

We thank the reviewer for this comment. We have added statements in the conclusion section for suggestions for further clinical practice of the results. These statements read as: “Thus, our findings may provide valuable information for clinicians to alert middle-aged people regarding the increased risk of CVD.” (Lines 319-320).

Comment#10

Some references should be updated.

Response#:10

In response to the reviewer’s suggestion, we have replaced some old references, those references are in number 11,18,21,22, and add extra reference for citation, those reference are in 25,16,27,28. Accordingly, we have changed the sequence of citations in the text.

Reviewer 2

Comment:

The authors aimed to evaluate the walking speed for predicting cardiovascular disease (CVD) risk in middle-aged and elderly people. I am interested in this topic, however, the data is mainly collected from a specific community. Why did the authors choose the residents living in Guishan district to be enrolled in their study? Can this small group represent and draw conclusion in whole middle-aged and elderly population of Taiwan including the downtown, suburban, and rural areas?

Response#:

We appreciate the reviewer’s concern. This is a community-based study, so we chose the participants living in Guishan district for convenient and selective sampling. The resident from this district might have different characteristics comparing with the general population, and we only recruit participants from this district in northern Taiwan as our favor population. Thus, the results can not represent the other regions of Taiwan. Future studies using random sampling of communities with wider range of area would make the research more robust. To response to the reviewer’s concern, this limitation has been acknowledged in the discussion section, which read as “However, our study still had limitations. The participants in our study were recruited from a relatively small group in northern Taiwan as our favored population. The characteristics of the residents from this small group might differ from characteristics in the general population, so the findings cannot be generalized to the whole middle-aged and elderly population in Taiwan. Selection bias should be considered, and the results of our study should not be extrapolated to other regions of Taiwan. Future studies using random sampling of communities with a wider range of regions would make the research more discursive.” (Lines 306-312).

General Comments:

I applaud the authors for their effort, however, the syntax, grammar and overall style of this manuscript is sometimes difficult to read and needs a robust editing from a fluent or native English speaker. For instance, in Introduction section, last sentence: “and elderly poeple.” is an obvious typo error. I suggest the authors to have the manuscript carefully proofread again to correct grammar and other errors.

Response to General Comments:

We thank the reviewer for reminding us this important issue. We had found a native English speaker to help us to edit our manuscript before we resubmitted it, and the English editing certificate was provided. (Please see Appendix 2)

Major Comment 1:

Please revise the title of your manuscript so that it contains details of the study design which characterize the investigation as well. Please provide more information on the novelty of your study. Introduction should finish with the aim of the study that should be described more clearly.

Response to Major Comment 1:

We appreciate the suggestion form reviewer. After we revised our manuscript, the title was rewrote as suggested in cover page. The title now reads as: “The Association between Walking Speed and Risk of Cardiovascular Disease in Middle-aged and Elderly people in Taiwan, a community-based, cross-sectional study”.

We fully agree with the reviewer’s comment about novelty finding of our study, we have added statements in discussion, which read as: “Previous research has mainly focused on the relationship between walking speed and CVD risk [25] [26] [27] [28] in other countries. Our research was the first community-based study that examined the role of age in the relationship between walking speed and CVD risk in Taiwan. Thus, the novel findings of this study are the first to explore the association between CVD risk and walking speed in different age populations in Taiwan. Second, we conducted a community-based study and comprehensively collected various data from a health promotion project, which may have clinical implications.” (Lines 299-305).

According reviewer’s helpful suggestion, we also modified our aim in the introduction, which read as “However, previous research has mainly focused on the relationship between walking speed and CVD risk only. Our research was the first community-based study that aimed to reveal the role of age in the relationship of walking speed and CVD risk.” (lines 72-75)

Major Comment 2:

There were total 400 participants in the project, which was a relatively small sample size. Sample size should be increased to claim scientific merits of your study. The claims made by authors are significantly weakened by the fact that they are talking about findings drawn from a small sample size, and then should have mentioned this as limitation without any justification. Did you calculate the sample size before the investigation and what should be the minimal sample size needed in this study to avoid underpowered? Could the authors show the power of their sample size?

Response to Major Comment 2:

We fully agree with the reviewer regarding the fact that our findings were obtained from community-based subjects and cannot be generalized to the whole middle-aged and elderly population in Taiwan. To response to the reviewer’s comment, this limitation has been acknowledged in the discussion section, which read as “However, our study still had limitations. The participants in our study were recruited from a relatively small group in northern Taiwan as our favored population. The characteristics of the residents from this small group might differ from characteristics in the general population, so the findings cannot be generalized to the whole middle-aged and elderly population in Taiwan. Selection bias should be considered, and the results of our study should not be extrapolated to other regions of Taiwan. Future studies using random sampling of communities with a wider range of regions would make the research more discursive.” (Lines 306-312). We also calculated sample size with G*power 3.1 software before this study. We did calculate the minimum sample size. We also added statements to explain the calculation in study design and participants. These statements read as: “Before this study, the minimum sample size for this study was calculated at the initial stage of the study after previewing a relatively smaller population. Considering an 85% power, a 95% confidence level, and 0.30 as the high CVD risk rate among middle-aged people, we calculated that 288 participants were required to detect differences between these two study groups with an odds ratio of at least 2.” (Lines 79-83).

1. Jacob C, Statistical power analysis for the behavioral sciences. International Biometric Society, 1988.

Major Comment 3:

Please indicate the selection of study participants by adding a CONSORT diagram with flow chart.

Response# Major Comment 3:

Thanks for the reviewer’s suggestion, we made the flow chart in response to this comment (Please see Appendix 1) ,and put it in manuscript as Figure 1.

Major Comment 4:

The Introduction and Discussion sections need to be expanded with improvements. Try to cite more references to support your Result. Moreover, some of the references are outdated. Please update your reference list.

Response# Major Comment 4:

In response to the reviewer’s suggestion, we add material in introduction, which read as” However, previous research has mainly focused on the relationship between walking speed and CVD risk only. Our research was the first community-based study that aimed to reveal the role of age in the relationship of walking speed and CVD risk.” (Line 72-75), we add material in discussion, which read as” Previous research has mainly focused on the relationship between walking speed and CVD risk [25] [26] [27] [28] in other countries. Our research was the first community-based study that examined the role of age in the relationship between walking speed and CVD risk in Taiwan. Thus, the novel findings of this study are the first to explore the association between CVD risk and walking speed in different age populations in Taiwan. Second, we conducted a community-based study and comprehensively collected various data from a health promotion project, which may have clinical implications. However, our study still had limitations. The participants in our study were recruited from a relatively small group in northern Taiwan as our favored population. The characteristics of the residents from this small group might differ from characteristics in the general population, so the findings cannot be generalized to the whole middle-aged and elderly population in Taiwan. Selection bias should be considered, and the results of our study should not be extrapolated to other regions of Taiwan. Future studies using random sampling of communities with a wider range of regions would make the research more discursive.” (Line 299-305, 306-312) We also have replaced some old references, those references are in number 11,18,21,22, and add extra reference for citation, those reference are in 25,16,27,28. Accordingly, we have changed the sequence of citations in the text.

Appendix 1.

<please see the file "response 2 edited">

Appendix 2.

<please see the file "response 2 edited">

---

## [Decision Letter · Decision Letter 1]

12 Jun 2020

The Association between Walking Speed and Risk of Cardiovascular Disease in Middle-aged and Elderly people in Taiwan, a community-based, cross-sectional study

PONE-D-20-04269R1

Dear Dr. Chen,

We’re pleased to inform you that your manuscript has been judged scientifically suitable for publication and will be formally accepted for publication once it meets all outstanding technical requirements.

Kind regards,

Yu Ru Kou, PhD

Academic Editor

PLOS ONE

Additional Editor Comments (optional):

Reviewers' comments:

Reviewer's Responses to Questions

**Comments to the Author**

1. If the authors have adequately addressed your comments raised in a previous round of review and you feel that this manuscript is now acceptable for publication, you may indicate that here to bypass the “Comments to the Author” section, enter your conflict of interest statement in the “Confidential to Editor” section, and submit your "Accept" recommendation.

Reviewer #1: All comments have been addressed

Reviewer #2: All comments have been addressed

2. Is the manuscript technically sound, and do the data support the conclusions?

Reviewer #1: Yes

Reviewer #2: Yes

3. Has the statistical analysis been performed appropriately and rigorously? 

Reviewer #1: Yes

Reviewer #2: Yes

4. Have the authors made all data underlying the findings in their manuscript fully available?

Reviewer #1: Yes

Reviewer #2: Yes

5. Is the manuscript presented in an intelligible fashion and written in standard English?

Reviewer #1: Yes

Reviewer #2: Yes

6. Review Comments to the Author

Reviewer #1: The reviewer's comments have been adequately addressed. I am pleased to accept the revised version. I have no further comments.

Reviewer #2: Thanks for all answers and changes in the revised manuscript. The authors addressed properly to my previous comments and reviewed in-depth the manuscript. I have no further comments and I recommend publication of the paper in its current form.

7. PLOS authors have the option to publish the peer review history of their article (what does this mean?). If published, this will include your full peer review and any attached files.

Reviewer #1: No

Reviewer #2: No

---

## [Editor Report · Acceptance letter]

19 Jun 2020

PONE-D-20-04269R1 

The Association between Walking Speed and Risk of Cardiovascular Disease in Middle-aged and Elderly people in Taiwan, a community-based, cross-sectional study 

Dear Dr. Chen:

I'm pleased to inform you that your manuscript has been deemed suitable for publication in PLOS ONE. Congratulations! Your manuscript is now with our production department. 

Kind regards, 

on behalf of

Dr. Yu Ru Kou 

Academic Editor

PLOS ONE